# ARTEMIS: Detecting Airdrop Hunters in NFT Markets with a Graph Learning System

## ABSTRACT

As Web3 projects leverage airdrops to incentivize participation, airdrop hunters tactically amass wallet addresses to capitalize on token giveaways. This poses challenges to the decentralization goal. Current detection approaches tailored for cryptocurrencies overlook non-fungible tokens (NFTs) nuances. We introduce ARTEMIS, an optimized graph neural network system for identifying airdrop hunters in NFT transactions. ARTEMIS captures NFT airdrop hunters through: (1) a multimodal module extracting visual and textual insights from NFT metadata using Transformer models; (2) a tailored node aggregation function chaining NFT transaction sequences, retaining behavioral insights; (3) engineered features based on market manipulation theories detecting anomalous trading. Evaluated on decentralized exchange Blur's data, ARTEMIS significantly outperforms baselines in pinpointing hunters. This pioneering computational solution for an emergent Web3 phenomenon has broad applicability for blockchain anomaly detection.

## CCS CONCEPTS

• **Do Not Use This Code → Generate the Correct Terms for Your Paper**; *Generate the Correct Terms for Your Paper*; Generate the Correct Terms for Your Paper; Generate the Correct Terms for Your Paper.

## KEYWORDS

Airdrop hunters, Web3, NFTs, Graph neural network, Multimodal deep learning

**ACM Reference Format:**
Anonymous Author(s). 2018. ARTEMIS: Detecting Airdrop Hunters in NFT Markets with a Graph Learning System. In *Proceedings of Make sure to enter the correct conference title from your rights confirmation emai (Conference acronym 'XX)*. ACM, New York, NY, USA, 10 pages. https://doi.org/XXXXXXX.XXXXXXX

## 1 INTRODUCTION

The practice of airdrops has become commonplace amongst Web3 business operations, with Decentralized Applications (DApps) seeking to incentivize widespread user participation in their interactions and activities through targeted dispensation of tokens, governed by predetermined rules in smart contracts [22]. However, a corresponding cohort rising in the Web3 realm are "airdrop hunters','

who actively amass swathes of wallet addresses to interact with relevant smart contracts and capitalize on these bountiful token giveaways [8]. As DApps increasingly utilize airdrops as rewards for early adopters, the hunters' strategies could net them hefty surging gains, though simultaneously pose challenges to DApps' decentralization ambitions. Specifically, with intricate on-chain behaviors involving self-trading across their manifold holdings to mimic organic transactions, airdrop hunters cunningly exploit DApps' largesse towards ostensibly active participants, which has been critiqued due to threaten ecosystem vital for DApps' functioning [20]. Correspondingly, the onus falls on the team behind the DApps to institute checks against siphoning of incentives by those airdrop hunters, without penalizing genuinely engaged users.

Although airdrops and the corresponding hunters represent an emerging business model and community, relevant research remains scarce. Fan et al. [8]'s research demonstrates identifiable and observable patterns among airdrop hunters' address activities. The simplest example is "transaction loops" cycling assets between their wallets to mimic exchanges. But such straightforward techniques often get flagged by DApps' monitoring systems, prompting airdrop hunters to evolve more sophisticated strategies [1]. This illuminates the limitations of visualizing wallet interactions to detect increasingly complex fraud, falling short of required responsiveness. Moreover, current studies mostly focus on cryptocurrency and ignore the airdrop hunter issue in the NFT context.

Currently, there have been some attempts based on machine learning to detect fraud behaviors on the blockchain. Among these works, graph-based modeling of wallet interactions is a very intuitive approach and has produced many effective detection frameworks for phishing scams [31], money laundering [21], and bot arbitrage [13]. Consequently, constructing airdrop hunter detection models using machine learning based explicitly on a graphic way is logical. These works offer valuable references for developing our airdrop hunter detection system, but directly adopting them has limitations. Specifically,

**(i)** Existing GNN modeling methods cannot accurately characterize transaction paths. Merging multiple edges between the same node pairs in the graph discards critical sequencing data for current airdrop hunter detection. **(ii)** Related works lack the utilization of intrinsic NFT features. Current practices only consider homogeneous cryptocurrency transactions, not accounting for additional information tied to NFTs as traded assets. **(iii)** Absence of tailored feature engineering. With a focus on tracing airdrop hunters in NFT transactions, factors like NFT heterogeneity introduce more noise. More sophisticated feature extraction could bolster modeling effectiveness amidst such intricacies.

Our primary focus revolves around tracking Airdrop Hunters in NFT transactions, a prevalent trading scenario in Web3. Addressing this, we introduce ARTEMIS: AiRdrop hunTErs detection via a

MultImodal and graph learning System. In response to the aforementioned limitations, this system presents three tailored solutions:

**(i)** We devise a tailored neighbor sampling method and aggregator that chains together multi-hop NFT transaction sequences, incorporating crucial behavioral information. **(ii)** We design multimodal feature extraction modules, leveraging Transformer-based pre-trained models to extract visual and textual insights from NFTs. **(iii)** We engineer common NFT price representations and advanced hunter-oriented features based on market manipulation theories and domain knowledge.

In summary, the contributions of this work are:

- We formalize the problem definition of airdrop hunter detection in the NFT market context, and label hunters within Blur marketplace data as a dataset.
- We propose the ARTEMIS, the first work attempting systematic airdrop hunter detection using the machine learning method. Our system significantly outperforms existing models for hunter identification. We also introduce tailored strategies during ARTEMIS training to address associated challenges effectively.
- We design multimodal feature extraction, transaction path-based multi-hop neighbor sampling and aggregation, and advanced feature representation modules. Experiments validate each component's utility. This modular system is transferable to other downstream tasks, with each module broadly applicable to other NFT or on-chain anomaly detections.

## 2 BACKGROUND AND RELATED WORKS

### 2.1 Blur and Airdrop Hunters

Unlike the read-only Web1 and platform-controlled Web2, Web3 leverages blockchain technologies like smart contracts and cryptocurrencies to put asset ownership back into users' hands [27]. As a vital Web3 application, non-fungible tokens (NFTs) are a new form of digital asset, each representing a unique artwork, certificate, etc., unlike traditional cryptocurrencies such as Bitcoin and Ethereum [11]. Attributing to these traits, the NFT market exploded in 2021, with total market value surging to around $10 billion by early 2023 [25]. In the NFT landscape, decentralized exchanges operating via smart contracts are crucial for bolstering market liquidity and ecosystem growth, which has long been dominated by OpenSea[1] through first-mover advantage. Blur[2] entered the NFT market as an aggregator platform in Oct. 2022, relatively late but empowered by three rounds of token airdrops to incentivize participants. During Blur's second airdrop on Feb. 15th, 2023, over 300 million tokens were distributed (over 10% of the total supply), drawing 115,834 users to surpass OpenSea [32].

Airdrops are a common token distribution approach utilized by numerous Web3 projects like Convex and AAVE, whereby tokens are allocated to users at launch per set criteria to foster long-term holdings or activity [2]. Post-airdrop, Blur's daily active users exploded and then steadily climbed, affirming the immense potential of this Web 3 growth strategy (Fig. 1). However, this incentive structure predictably attracted copious airdrop hunters. Analysts reveal

[1]https://opensea.io/
[2]https://blur.io/

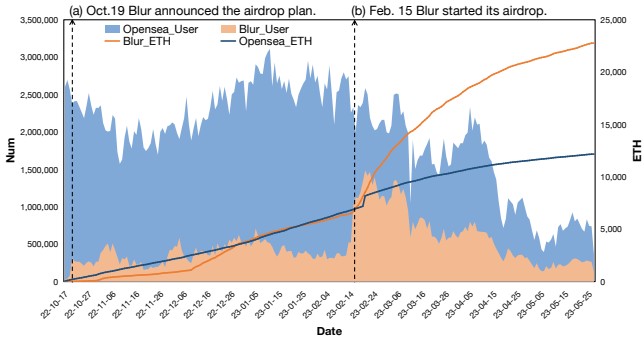

Figure 1: As a late entrant to the market, Blur announced on Oct. 19, 2022, that it would adopt an airdrop strategy, subsequently attracting users and transactions. On Feb. 15, 2023, Blur commenced its airdrop distribution, spurring a surge in daily active users and trade volumes eclipsed OpenSea's.

that 50% of Blur's NFT trading volume derives from less than 300 wallets, while 1% of "whales" hold 84% of total value locked in Blur's bid pools [23]. This implies rampant wash trading on Blur, where a traded NFT's buyer and seller are the same airdrop hunter. Such behavior stifles platform growth and triggers market contagion amidst NFTs, jeopardizing overall market health.

### 2.2 Graph Representation Learning on Blockchain

Recently, the integration of blockchain and machine learning has garnered a plethora of notable research. This convergence becomes especially pivotal in scenarios such as anti-money laundering, phishing scam detection, and de-anonymization. Given that wallet interactions on the blockchain inherently form a network structure, it offers an ideal landscape for graph representation learning.

In the random walk-based sequence generation, though DeepWalk[24] stands as a hallmark, several advancements have also emerged. Wu et al. devised Trans2Vec[30], integrating transaction timestamps and amounts into a biased random walk process, aiming to capture transaction relationships between nodes more authentically. In a similar vein, Lin et al. embarked on a time-weighted random walk approach[17, 18]. Venturing further, Hu et al. considered the heterogeneity of nodes and introduced a "Jump-Stay" temporal-weighted biased walking method[12] for heterogeneous multi-graph modeling, balancing the distribution of diverse node types.

In the domain of GNN, the Graph Convolutional Network (GCN) is a prominent representative [16]. Shen et al., for example, successfully applied GCN to phishing detection tasks in the blockchain context[26]. Pushing the envelope, Zhou et al. incorporated attention mechanisms, proposing a Hierarchical Graph Attention Network for account de-anonymization challenges[33]. Moreover, Lo et al. unveiled Inspection-L[21], an innovative self-supervised GNN node embedding framework, which achieved state-of-the-art results on the Elliptic money laundering detection dataset. Kanezashi et al., focusing on the heterogeneity of nodes, adopted heterogeneous modeling and conducted an exhaustive evaluation of multiple GNN performances[14].

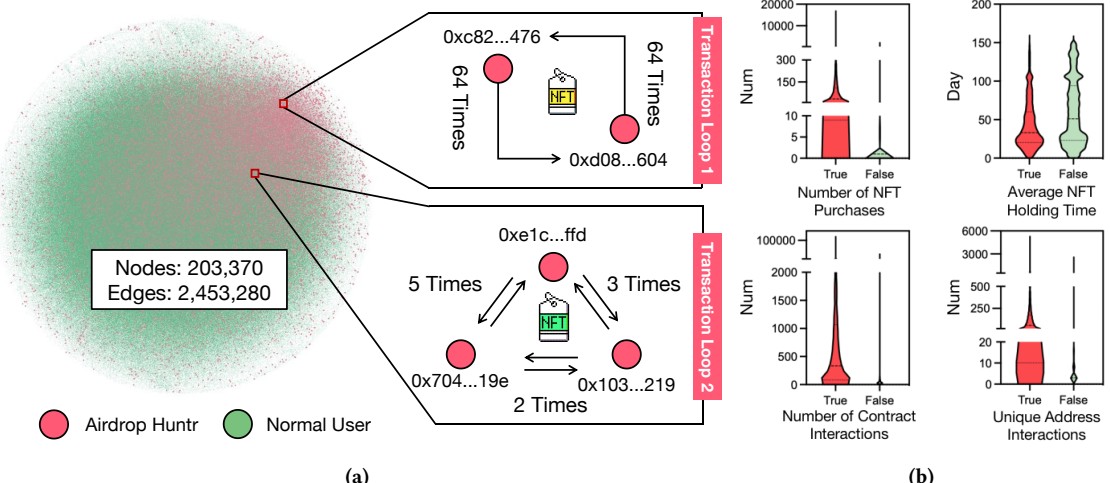

Figure 2: (a) The overview of our dataset. (b) The comparison of airdrop hunters and normal users.

## 2.3 Transformer Pre-trained Models

Over the past few years, pre-trained models have made significant strides, particularly in the fields of Computer Vision (CV) and Natural Language Processing (NLP). Many pivotal advancements in these domains have been achieved through the construction and optimization of pre-trained Transformer models. In the CV arena, the Vision Transformer (ViT) proposed by Dosovitskiy and colleagues leverages the self-attention mechanism of Transformers, demonstrating performance on par with or even surpassing traditional Convolutional Neural Networks[7]. Following closely, Caron and team introduced DINO[3], which is capable of learning visual representations without labels, further propelling the progress in self-supervised learning. Concurrently, in the NLP sphere, the BERT model[5], introduced by Devlin and associates in 2019, utilizes bidirectional Transformers to pre-train extensive text data, offering robust representational learning for downstream tasks. The RoBERTa[19] model is a robustly optimized version of BERT that enhances performance and universality by tweaking BERT's training strategy and data processing workflow. The emergence of these models has enriched the pre-trained resources available for research in the CV and NLP fields, facilitating the evolution of various applications.

Innovative works have pioneered the application of pre-trained models to blockchain-centric tasks, achieving some breakthroughs. For instance, the BERT4ETH[13] model aims to utilize a pre-trained Transformer to detect fraudulent activities on Ethereum, showing significant advantages. In predicting the selling price of NFT, the MERLIN framework[4], employing multimodal deep learning, exhibits remarkable predictive performance. Furthermore, in the realm of smart contract security auditing, research indicates that large language models like GPT-4 and Claude can identify contract vulnerabilities to a certain extent, albeit manual auditors are still required to mitigate false positive rates. These studies unveil the potential of pre-trained models in blockchain applications.

## 3 DATASET

We first elucidate the construction and processing of our dataset. After defining airdrop hunter detection as our initial research objective, we successfully compiled transaction data from the NFT marketplace, Blur, over a designated period and annotated associated addresses.

**Data Collection.** We utilized the Etherscan API[3] to compile all NFT transaction data and airdrop records related to Blur from Oct. 19, 2022, to Apr. 1, 2023. For traded NFTs, we thoroughly collected metadata, including NFT images, descriptions, and attributes. Adopting previous works' methodology, we leveraged clustering techniques to process transaction information. Through subsequent labeling, we compared airdrop records to identify airdrop hunters meticulously. Subsequently, we sampled varying hunter scales and visualized microscopic transaction paths to validate data reliability.

**Data Description.** Across the Blur marketplace, we acquired 2,453,280 NFT transactions encompassing 203,370 unique user wallet addresses. Total airdrops from Blur's official address[4] were 123,815. Of all receiving wallets, 4,808 (about 4% of total address) were labeled airdrop hunters, the rest regular traders. We logged timestamps, type (buy or sell), value (based on ETH token), sending/receiving addresses, NFT collection, and relevant NFT ID for each transaction. For every wallet, we compiled entire historical transaction and smart contract interaction records. For NFTs themselves, we gathered full metadata for 1,155,947 traded tokens. Fig. 2a shows the overview of our dataset. Simultaneously, we display two simplified real-world examples: In transaction loop 1, two addresses traded the same NFT back and forth 64 times. In loop 2, three addresses reciprocally exchanged a single NFT 10 times.

**Statistical Analysis.** We statistically analyzed several characteristic features between the labeled airdrop hunter (in the following work, we describe them with the label of "True") and normal node

---

[3]https://etherscan.io/
[4]0xf2d15c0a89428c9251d71a0e29b39ff1e86bce25

(comparatively, labeled with "False") groups. Fig. 2b shows some features between these two user clusters. 1. While both populations contained outliers with high NFT purchases, hunter's extremes were more pronounced. The hunter distribution also clearly exceeded regular users. Against information asymmetry in NFT markets, this purchase significance implies hunters are not motivated by specific NFTs. 2. For average hold times post-purchase, hunters were shorter (36 days) than regulars (53 days), fitting profit-driven aims to maximize airdrops via trading. 3. Hunter's interactions with the smart contract also had higher extremes and means than regular users. 4. Finally, we describe the distribution of unique addresses for each address transacted within these two clusters, another metric where hunters significantly diverged from regular users. These observations reinforce our hypothesis that airdrop hunters exhibit apparent behavioral differences from normal participants, which provide solid foundations for a GNN to learn and detect hunters' multidimensional feature patterns.

**Brief Conclusion**. With Blur's initial lenient airdrop rules and lack of hunter detection mechanisms, we observed hunters routinely employ transaction loops to inflate activity for airdrop eligibility artificially. However, as Blur refined its airdrop policies and instituted logic to deter these basic tactics, hunters had to adopt more intricate strategies. Similarly to other Web3 projects, comprehensive detection via conventional means (e.g., rules-based filtering on structural features) becomes very challenging in this situation [8]. Nonetheless, given hunters' consistent underlying motivation to maximize their airdrop acquisition, we posit that multi-dimensional analysis of address attributes, transaction patterns, and traded asset (in our study, the NFT) characteristics using GNNs may uncover unique collective on-chain behavior to identify hunters amidst complexity effectively.

## 4 MOTIVATION

Our work is the first to systematically detect airdrop hunters using graph-based machine learning in NFT trading contexts. Several critical insights motivated our design:

i. Graph Representations of Blockchain: Previous blockchain graph modeling traded off between GNNs and random walks. While GNNs excel at feature representations, they poorly retain transaction sequence information, conflicting with our aim to identify hunters via trade paths. In contrast, earlier random walk studies attempted to integrate sequencing through refined walk algorithms but sacrificed GNNs' powerful modeling capabilities. For our application, we desire both strengths. Specifically, capitalizing on NFT traceability, we prioritize sequential neighbors during graph neighbor sampling.

ii. Unique NFT Attributes: NFT heterogeneity presents opportunities for more discerning models. Intuitively, high-quality NFT records are more reliable, while hunters may manipulate low-quality ones. We posit NFT visual and textual traits as critical for assessing value and incorporate NFT feature extraction to combine quality cues with other signals to evaluate transaction legitimacy.

iii. Advanced Pricing Features: Unlike fungible tokens, each NFT has unique pricing, complicating pattern detection

from transaction values. Therefore, more sophisticated characteristics are needed to capture market manipulation traits accurately. We referred to Benford's law and the roundness detection of transaction tail numbers, which is widely used in market manipulation detection [6], to extract higher-order features from transaction prices to determine whether a transaction occurred "naturally" or by hunters.

In summary, by holistically synthesizing blockchain technologies, NFT features, and user behavioral perspectives into a tailored detection system, we aim to uncover the subtle signals distinguishing sincere participants from airdrop hunters.

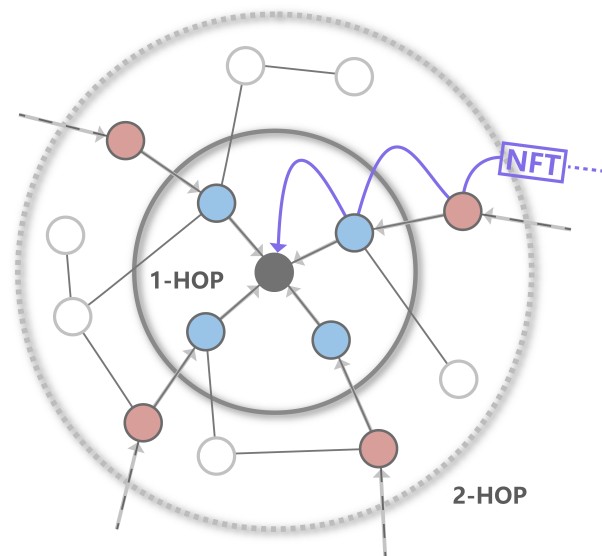

**Figure 3: Neighbor sampling based on transaction paths. Blue nodes are randomly sampled 1-hop nodes that have direct NFT transactions with the center node. Red nodes are 2-hop nodes that trace the corresponding NFT transaction paths. This process can be extended to K depth.**

## 5 ARTEMIS

In addressing the task of detecting Airdrop Hunters, we propose ARTEMIS: AiRdrop hunTErs detection via a MultImodal and graph learning System. In this section, we will elucidate our design rationale and introduce the various modules of ARTEMIS.

### 5.1 Graph Sampling and Aggregation

In this subsection, we describe the core module within ARTEMIS, which entails an enhancement of the aggregation function in graph neural networks. We leverage the transaction paths of NFTs as a guide for neighbor sampling and node information aggregation. Unlike random sampling, our algorithm prioritizes sampling along the NFT transaction paths, ensuring that the generated embeddings can capture the context of transactions, and obtain ample information. This sampling algorithm aligns with our design philosophy of characterizing node embeddings through transaction paths.

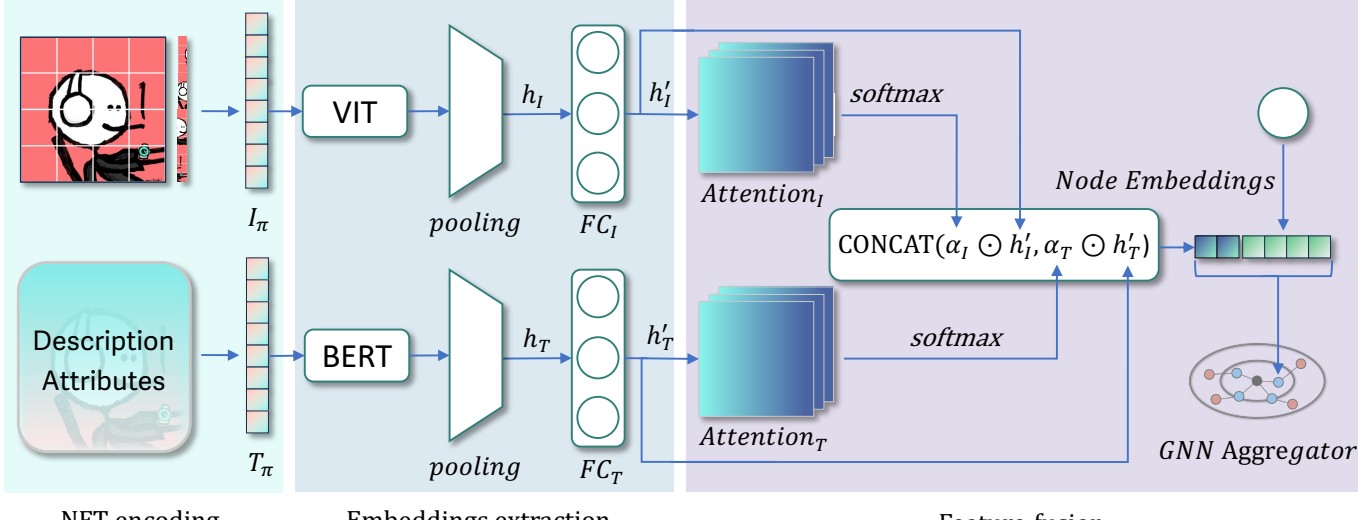

**Figure 4: NFT multimodal feature extraction module. Fine-tune pretrained BERT and ViT to generate embeddings from NFT's image and text descriptions. Then fusioned embeddings are involved in the training of downstream graph model**

*5.1.1* **Neighbour Sampling.** Define a graph $G = (V, E)$, where nodes $V$ represent wallet addresses of transactions, and $E$ represents NFT transactions. Our forward propagation algorithm generates embeddings for each node. We assume the model has been pre-trained, with fixed parameters including the aggregation function and weight matrices. At each depth, nodes aggregate information from their neighbours. For the first hop, any neighbour can be sampled. However, for subsequent hops, the sampling is based on the transaction paths. For instance, if node $V_0$ (central node) transacted $NFT_a$ with node $V_1$ (one-hop neighbour), then while sampling two-hop neighbours for $V_0$, with $V_1$ as the intermediate node, only nodes that transacted $NFT_a$ are sampled. This ensures that the sampled neighbourhood is not merely a random subset, but a meaningful set of nodes sharing the same NFT transaction history. Fig. 6 shows a detailed illustration of the process, which can be extended to multi-hop neighbors.

*5.1.2* **Embedding Generation.** Upon completing the neighbour sampling, each node updates its representation not only using its own current representation but also incorporating information from its neighbours. To achieve this, we concatenate the current representation of the node with the aggregated vector from its neighbourhood. This concatenated vector is then passed through a weight matrix for a linear transformation, followed by a nonlinear activation function $\sigma$, such as ReLU, to obtain the new node representation. In our subsequent work, we further incorporate NFT features into this node representation. These NFT features are derived from our multimodal feature extraction module, with more details to be discussed in the next subsection.

*5.1.3* **Neighbourhood Definition and Computation Strategy.** To ensure computational efficiency and consistency across each

batch processing, we adopt a fixed-size strategy during neighbourhood sampling. Our method always samples a fixed-size neighbourhood for each node. Specifically, for any node $v$, its neighbourhood $N(v)$ is defined as a fixed-size subset obtained by uniformly sampling from the set of nodes connected to it, denoted as $\{u \in V : (u, v) \in E\}$. In each iteration of forward propagation, we perform such uniform neighbourhood sampling anew for every node.

## 5.2 NFT Multimodal Feature Extraction

We introduce the NFT Multimodal Feature Extraction module within ARTEMIS. As the transaction targets for Airdrop Hunters, each NFT carries its unique image and text description. We utilize pre-trained vision models (ViT) and pre-trained language models (BERT) to extract visual and textual features from NFTs, respectively. In subsequent stages, we fuse these two types of representations to generate a unified embedding, which then participates in the graph model's aggregation module.

**Input Data Representation.** Consider a series of NFT datasets $\mathcal{D}$, where each data object $\pi \in \mathcal{D}$ is composed of a pair: an image $I$ and its corresponding text description $T$. Both the image and text are initially transformed into token sequences, represented as:

$$I_\pi = [i_1, i_2, \ldots, i_n] \tag{1}$$

$$T_\pi = [t_1, t_2, \ldots, t_n] \tag{2}$$

**Text Learning.** For a given NFT $\pi$, we employ a Transformer-based Pre-trained Language Model (PLM) – BERT, to perform deep contextualization of the token sequence of the text part $T$, mapping it to a $d_T$-dimensional space:

$$\text{Embed}_T = \text{BERT}(T_\pi) \tag{3}$$

where $\text{Embed}_T \in \mathbb{R}^{m \times d_T}$.

Using a pooling function, we obtain an embedding vector representing the entire text:

$$h_T = \text{pooling}(\text{Embed}_T) \quad (4)$$

where $h_T \in \mathbb{R}^{d_T}$. In this context, the polling function produces a special token, [CLS], which represents the entirety of the input. The same is true for the pooling function that follows.

**Image Learning.** Similarly, for the image part of the NFT, we utilize a Transformer-based Pre-trained Vision Model (PVM) – ViT, for processing:

$$\text{Embed}_I = \text{ViT}(I_\pi) \quad (5)$$

where $\text{Embed}_I \in \mathbb{R}^{n \times d_I}$.

Using a pooling function, an embedding vector representing the entire image is obtained:

$$h_I = \text{pooling}(\text{Embed}_I) \quad (6)$$

where $h_I \in \mathbb{R}^{d_I}$.

**Feature Fusion.** To obtain a fused representation of both text and image for each NFT, we first pass $h_T$ and $h_I$ through two different fully connected layers for dimensionality reduction:

$$h'_T = FC_T(h_T) \quad (7)$$

$$h'_I = FC_I(h_I) \quad (8)$$

where $h'_T \in \mathbb{R}^{d'_T}$ and $h'_I \in \mathbb{R}^{d'_I}$.

Subsequently, we utilize a self-attention mechanism to compute the weights of these two embeddings:

$$\alpha_T = \text{softmax}(Attention(h'_T)) \quad (9)$$

$$\alpha_I = \text{softmax}(Attention(h'_I)) \quad (10)$$

The final fused representation consists of a concatenation of the weighted embeddings:

$$h_A = \text{concat}(\alpha_T \odot h'_T, \alpha_I \odot h'_I) \quad (11)$$

where $h_A \in \mathbb{R}^{d'_T + d'_I}$.

Subsequently, the NFT embeddings are concatenated with the address embeddings and are utilized in the downstream graph neural network training. For the complete embeddings aggregation learning process, please refer to sections 5.1 and 5.2, and see Algorithm 1 for details.

## 5.3 Advanced Features

To address the practical task of Airdrop Hunters Detection with a targeted approach, this section elaborates on the effective features we constructed during the modeling process, along with our insights and some tests regarding these features.

Market Manipulation Price Features.Each NFT carries a unique value associated with it, posing a challenge for the model to extract generalizable information, especially from the prices of NFTs as it's hard for the model to directly learn potential market patterns, necessitating more sophisticated feature extraction techniques. We hypothesize that the activities of Airdrop Hunters are essentially market manipulation behaviors and validated this using two tests: Benford's Law and the rounding test of transaction prices (see Appendix for test results). Benford's Law utilizes the leading digit of price datasets to detect market manipulation. Specifically, the

---

**Algorithm 1** Sampling and Aggregation by Transaction Paths

**Require:** Graph $G(V, E)$ with edge attributes for NFTs
**Require:** Node features $\{x_v, \forall v \in V\}$
**Require:** Depth $K$
**Ensure:** Vector representations $Z = \{z_v, \forall v \in V\}$
1: **for** $k = 1$ to $K$ **do**
2:     **for all** $v$ in $V$ **do**
3:         **if** $k == 1$ **then**
4:             $N_v^k \leftarrow \text{inverse\_frequency\_sample}(G.\text{neighbors}(v))$
5:         **else**
6:             **for all** $u$ in $N_v^{k-1}$ **do**
7:                 $\text{NFT} \leftarrow \text{edge\_attribute}(G, v, u)$
8:                 $N_u^k \leftarrow \text{sample}(\{w|$
9:                 $w \in G.\text{neighbors}(u) \wedge$
10:                $\text{edge\_attribute}(G, u, w) = \text{NFT}\})$
11:         **end for**
12:         **end if**
13:         $h_v^k \leftarrow \text{AGGREGATE}(\{h_u^{k-1} \oplus h_{NFT}, \forall u \in N_v^k\})$
14:         $h_v^k \leftarrow \sigma(W^k \times \text{CONCAT}(h_v^{k-1}, h_v^k))$
15:     **end for**
16: **end for**
17: **for all** $v$ in $V$ **do**
18:     $z_v \leftarrow h_v^K$
19: **end for**
20: **return** $Z$

---

probability of the leading digit $d$ (where $d \in \{1, 2, ..., 9\}$) should be given by the following formula:

$$P(d) = \log_{10}(d + 1) - \log_{10}(d) \quad (12)$$

Similarly, under market manipulation, certain trailing digits in prices may appear more frequently than would be expected in a random distribution. Inspired by these theories, we extracted the leading and trailing non-zero digits of prices as features to characterize the naturalness of transactions.

5.3.1 **Asset Turnover Features.** Through our observation of the simplistic strategy "transaction loop" previously, the trading strategies of Airdrop Hunters imply that their wallets often have higher asset turnover rates and multiple buyback behaviors. We extracted the average holding duration of NFT assets for each wallet, and those NFTs still held are calculated based on the time from purchase to the present. Similarly, we counted the average holding occurrences for each wallet concerning NFT assets.

5.3.2 **Wallet Activity Features.** The number of interactive addresses can help us understand the activity level of a wallet and its connections with other users. The ratio of transaction count to interactive address count reveals the transaction exclusivity of the wallet, and often, multiple wallets held by Airdrop Hunters stand out in this metric. Due to the complex airdrop computation rules, interactions with contracts without generating transactions could also lead to airdrops, hence we accounted for the number of contract calls for each wallet to augment the information.

5.3.3 **Acquisition of Airdrop Tokens.** This is a crucial post hoc feature with a direct correlation to whether an address belongs to

Airdrop Hunters. Airdrop Hunters employ a series of strategies with the explicit aim of acquiring airdrop tokens from events. It's imperative to note that our aim is to construct a real-time model for detecting Airdrop Hunters rather than post hoc inductions, therefore, the unannotated ARTEMIS in the subsequent experimental sections does not encapsulate this feature. We only mention and analyze this feature in the ablation study subsection, and conduct relevant analyses.

## 5.4 Training Strategies

In this subsection, we primarily introduce the training strategies tailored for ARTEMIS and explain the problems these strategies essentially address.

**Transaction Address Power Law Distribution.** The distribution of blockchain transaction addresses often follows a power-law distribution, meaning that a small number of high-frequency accounts appear massively in transactions. We tested the address distribution in the blur market and found it also follows a power-law distribution, with the results illustrated in fig. **??** (For test results see Appendix). From a graph construction perspective, this implies that a portion of nodes act as super-nodes, possessing a multitude of edges. These super-nodes, during training, can affect the feature representations of other nodes.

To mitigate this impact during training, we employed the following two measures:

**Inverse Frequency Sampling.** We aim to reduce the probability of sampling super-nodes during neighbor sampling to ensure effective learning. Since the 2-hop and beyond neighbor sampling is based on NFT transaction paths, here we only consider the initial neighbor sampling. We calculate the degree for each node's neighbors: degree($V_i$), rank the neighbor nodes in ascending order based on their degrees $r(V_i)$, and then compute the sampling probability:

$$P_{\text{sample}}(N_i) = \frac{\exp(-\beta \cdot r(N_i))}{\sum_j \exp(-\beta \cdot r(N_j))} \quad (13)$$

Here, $\beta$ is a hyperparameter, and $j$ iterates over all neighbor nodes.

In this formula, we employ the exponential function to emphasize the sampling priority of nodes ranked higher (i.e., with smaller degrees). The hyperparameter $\beta$ determines the extent of this emphasis: a larger $\beta$ value will grant significantly higher sampling probabilities to the few nodes with the smallest degrees, while a smaller $\beta$ will lead to a smoother distribution. The impact of this hyperparameter on model performance will be discussed in the subsequent experimental sections.

**Batch Balance.** We adopt a fixed quantity of neighbor sampling for training to ensure a balanced number of positive and negative samples in each batch. Employing a fixed neighbor count aims to reduce computational load and alleviate the influence of super-nodes, while balancing samples between batches aims to mitigate biases brought about by dataset imbalance.

## 6 EXPERIMENTS

## 6.1 Experimental Setup

**Task Description.** Experiments are conducted on the dataset described in Section 3 with the objective of detecting Airdrop Hunters, formulated as a binary classification problem where Airdrop Hunters are considered as the positive class. For experimental purposes, the dataset is split into training and validation sets in a ratio of 8:2. The evaluation metrics adopted are Precision, Recall, and the F1 score. Briefly, Precision quantifies the proportion correctly predicted as the positive class, Recall depicts the proportion of actual positive class correctly predicted, while the F1 score is the harmonic mean of the two.

**Baselines.** In this experiment, the ARTEMIS model is utilized and compared against three types of baseline models:

- Methods on structured data such as SVM and LightGBM are noteworthy. Since these methods are incapable of utilizing edge information, classification is solely based on node features.
- Methods based on graph random walks like DeepWalk and Node2Vec, which take advantage of both the graph structure and node features.
- Methods based on Graph Neural Networks (GNNs) like GCN, GraphSAGE, and GAT. For these methods, the same features as ARTEMIS are used.

**Implementation.** For the implementation of ARTEMIS, the following settings are employed: The number of layers $k$ in the graph neural network is treated as an experimental variable. The neighbor sample size is set to 20. The batch size is configured to 256 with a dropout ratio of 20%. In the NFT feature extraction module, ViT-base (patch16-224) is used as the pre-trained visual model (PVM) and BERT-base-uncased is employed as the pre-trained language model (PLM). A 12-layer Transformer encoder is set up with the hidden layer size $d_I = d_T$ defaulted to 768, and 12 attention-heads are used.

For the setup of baseline models: For methods based on random walks (DeepWalk and Node2Vec), the number of walks is set to 10, the walk length to 20, and the context size to 5. For all methods based on Graph Neural Networks, the number of GNN layers is set to 3, the neighbor sample size to 20, the batch size to 256, and the dropout ratio to 20%.

## 6.2 Performance Comparison

Each experiment was conducted five times, retaining the result of the experiment with the best F1 score.

**Table 1: Comparison for Airdrop Hunters Detection**

| Method | Precision | Recall | F1 |
|---|---|---|---|
| SVM [28] | 0.522 | 0.471 | 0.495 |
| LightGBM [15] | 0.543 | 0.501 | 0.521 |
| DeepWalk [24] | 0.481 | 0.409 | 0.442 |
| Node2Vec [9] | 0.459 | 0.418 | 0.443 |
| GCN [16] | 0.504 | 0.480 | 0.492 |
| GraphSAGE [10] | 0.597 | 0.565 | 0.580 |
| GAT [29] | 0.528 | 0.494 | 0.511 |
| **ARTEMIS** | **0.710** | **0.729** | **0.719** |

In Table 1, we present a comprehensive performance comparison of our method with various baselines. The following observations can be made: Both DeepWalk and Node2Vec, which are based on

**Table 2: Evaluating Non-real-time Improvement**

| Method | Precision | Recall | F1 |
|---|---|---|---|
| ARTEMIS | 0.710 | 0.729 | 0.719 |
| w/ airdrop count | **0.729** | **0.737** | **0.739** |

**Table 3: Impact of Aggregation Depth K on ARTEMIS Performance**

| Method | Precision | Recall | F1 |
|---|---|---|---|
| ARTEMIS_1 | 0.626 | 0.643 | 0.634 |
| ARTEMIS_2 | 0.692 | 0.704 | 0.698 |
| **ARTEMIS_3** | **0.710** | **0.729** | **0.719** |
| ARTEMIS_4 | 0.702 | 0.715 | 0.709 |

**Table 4: Impact of Frequency Inverse Order Sampling on Model Performance**

| Method | Precision | Recall | F1 |
|---|---|---|---|
| **ARTEMIS($\beta$=1.0)** | **0.710** | **0.729** | **0.719** |
| ARTEMIS($\beta$=0.1) | 0.673 | 0.675 | 0.674 |
| ARTEMIS($\beta$=0.5) | 0.692 | 0.690 | 0.691 |
| ARTEMIS($\beta$=2.0) | 0.668 | 0.647 | 0.657 |

random walks, exhibit relatively lower performance with F1 scores of 0.459 and 0.443, respectively. Their performance falls behind traditional machine learning approaches such as LightGBM and SVM, suggesting a higher dependency on capturing address-related features for the task. Among GNN methods, GraphSAGE performs exceptionally well with an F1 score of 0.580, surpassing other graph-based methods like DeepWalk, Node2Vec, GCN, and GAT. ARTEMIS significantly outperforms all other methods, achieving the highest precision, recall, and F1 score. With an F1 score of 0.719, ARTEMIS outperforms the second-best GraphSAGE by 0.139.

As Table 2, the enhancement to the model imparted by the post-event feature Airdrop Count. Given that our objective is to devise a real-time model, this post-event feature is utilized here solely for comparison. The performance of our current model does not significantly fall short when juxtaposed with the state attained with post-event features, thus preliminarily affirming that our model satisfactorily fulfills the requirement of real-time operation.

Table 3 illustrates the impact of selecting the depth parameter $K$ for neighbor sampling and aggregation while keeping other parameters fixed. Notably, the model's performance shows an upward trend as the depth of neighbor aggregation increases from 1 to 3. However, when the depth reaches 4, there is a slight performance decrease. This suggests that most valuable information can be distilled within three layers of neighbor aggregation, and increasing K beyond 3 starts to introduce more noise into the model.

We conducted comparative tests on the impact of the hyperparameter $\beta$ in Frequency Inverse Order Sampling on model performance. The experimental results in Table 4 demonstrate that the choice of $\beta$ has a significant effect on model performance. When $\beta = 0.1$, the strategy degenerates into something approximating

**Table 5: Ablation Study for Different Modules**

| Method | Precision | Recall | F1 |
|---|---|---|---|
| ARTEMIS | 0.710 | 0.729 | 0.719 |
| Ablation study of NFT multimodal modules | | | |
| w/o fine-tuning | 0.693 | 0.680 | 0.686 |
| w/o Image Embeddings | 0.698 | 0.678 | 0.688 |
| w/o Text Embeddings | 0.702 | 0.681 | 0.691 |
| w/o NFT Multimodal | 0.685 | 0.677 | 0.681 |
| Ablation study of other modules | | | |
| w/o Adv. Features | 0.643 | 0.610 | 0.626 |
| w/o Trade Neighb. Aggr. | 0.624 | 0.609 | 0.616 |

random sampling. When $\beta = 1.0$, ARTEMIS achieves the best performance on all metrics. This observation highlights the advantage of moderate non-linearity in capturing the intrinsic structure of the data. Further comparison shows that either larger or smaller values of $\beta$ (such as $\beta = 2.0$ and $\beta = 0.1$) lead to a decline in overall performance. This might suggest that either overly aggressive or conservative non-linearity is not applicable on this dataset.

## 6.3 Ablation Study

Table 5 presents the ablation study results. Initially, we observe a positive contribution from each module to the experimental results, with a noticeable decline when any of them is removed. Secondly, we find that the neighbor aggregation based on transactions plays the most crucial role; the model's F1 score drops by about 0.1 when this module is omitted. The advanced features also significantly impact the model's performance, indicating a successful deep characterization of NFT transactions through these features. Furthermore, we demonstrate the effectiveness of the NFT feature extraction module under various conditions. There's a notable performance decline in scenarios where BERT and ViT are not fine-tuned. Among Image and Text Embeddings, the image information proves to be more critical, aligning with our intuition that images hold more importance in NFTs.

## 7 CONCLUSION

This work represents the first step in building a deep learning system to detect airdrop hunters, a critical and emerging problem with implications for Web3 ecosystem health and future research directions of the WWW community. We formalize the novel task of airdrop hunter detection and benchmark the performance of baseline models. Through compiling on-chain data from NFT trading markets, we propose ARTEMIS, a multimodal graph neural network system tailored for this task. ARTEMIS contains three primary design modules and accompanying training strategies to address data distribution challenges. Subsequent experiments demonstrate the model's superiority, with ablation studies discussing each component's importance. Moreover, tracing NFT transaction paths and extracting multimodal NFT representations and generalized advanced features could transfer to other potential NFT-based machine-learning tasks. We provide one of the first specialized computational solutions for this frontier domain.

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

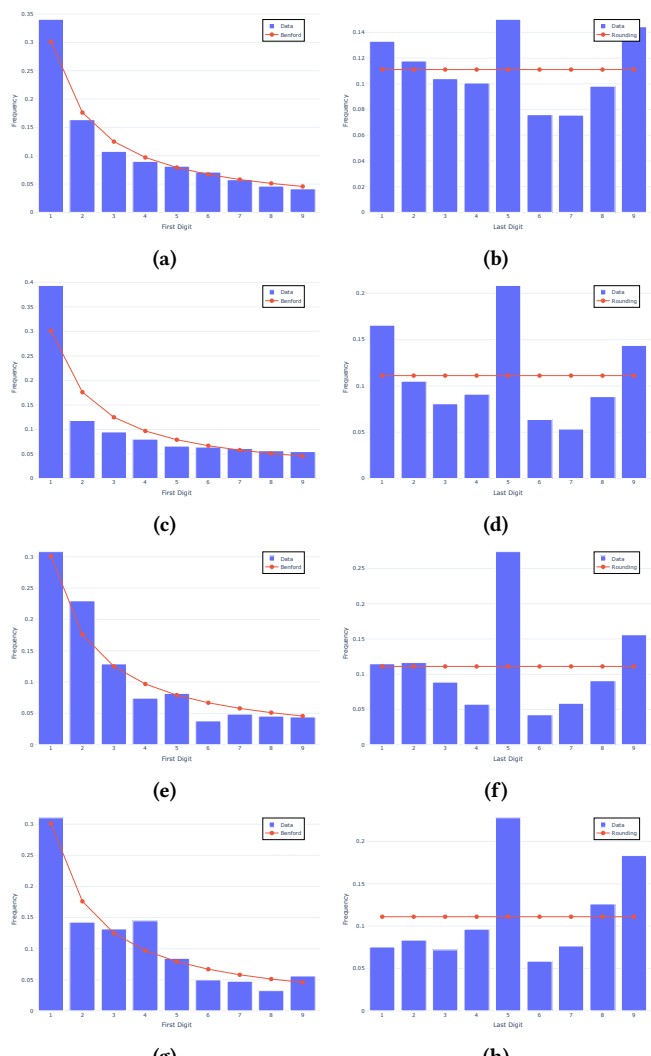

Figure 5: (a) Distribution of the first digits in NFT transaction prices on the Blur marketplace. (b) Distribution of the last digits in NFT transaction prices on the Blur marketplace. (c) Distribution of the first digits in NFT transaction prices on the Lookrare marketplace. (d) Distribution of the last digits in NFT transaction prices on the Lookrare marketplace. (e) Distribution of the first digits in NFT transaction prices on the Opensea marketplace. (f) Distribution of the last digits in NFT transaction prices on the Opensea marketplace. (g) Distribution of the first digits in NFT transaction prices on the X2Y2 marketplace. (h) Distribution of the last digits in NFT transaction prices on the X2Y2 marketplace.

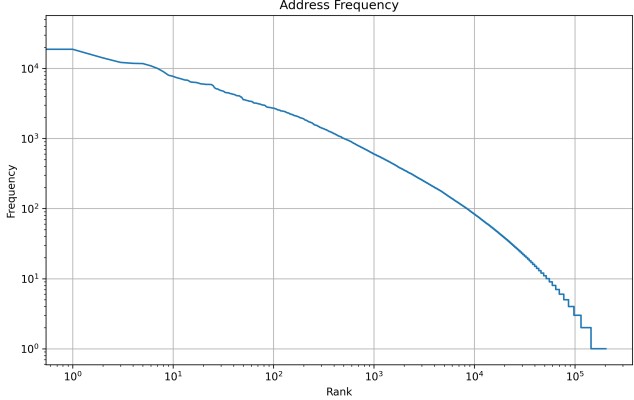

Figure 6: Address Power Law Distribution Test. The x-axis represents Rank while the y-axis represents Frequency, with both axes being on a logarithmic scale. When the frequency of the addresses and their ranks approximate a straight line on a log-log plot, it suggests that the distribution may follow a power law.

## A  APPENDIX

### A.1  Market manipulation detection

Benford's Law predicts the distribution of leading digits in many sets of numerical data. Instead of all digits (1 through 9) being equally likely, this law suggests that lower digits like 1, 2, or 3 are more likely to appear as the leading digit than higher digits like 8 or 9. It's often employed in forensic accounting and fraud detection, as significant deviations from the expected distribution may indicate manipulated numbers.

On the other hand, the Last Digit Rounding Law highlights the human tendency to round numbers, often leading to a disproportionate number of figures ending in specific digits, especially 0 or 5. Similar to Benford's Law, an unusually high occurrence of numbers ending in these rounded digits in financial or other data can hint at potential rounding or data manipulation.

The left half of Figure 5 shows the Benford's Law test for each market, while the right half displays the distribution of the last digit. The image indicates that the market distribution deviates somewhat from the expected distribution of Benford's Law. Additionally, the last digits are not as uniformly rounded as expected, suggesting a potential for market manipulation to some extent.

### A.2  Address frequency power-law detection

In Figure 6, we illustrate an empirical test to ascertain whether the distribution of addresses in the blockchain follows a power law, a common characteristic observed in various networked systems. The axes are plotted on a logarithmic scale to better discern the relation. The x-axis denotes the rank of addresses, which is determined by the frequency of their occurrences, while the y-axis represents the said frequency of occurrences. In a system following a power law distribution, a linear relationship is expected on a log-log plot, as exhibited by the data in the figure. This linear trend suggests that there are a few addresses (the "head" of the distribution) that occur very frequently, while the majority of addresses (the "tail") occur much less frequently. This distribution characteristic is crucial as it highlights the existence of 'hubs' or highly connected nodes, a feature common in many real-world networks.

Received 20 February 2007; revised 12 March 2009; accepted 5 June 2009

