# OpenReview forum: "ARTEMIS: Detecting Airdrop Hunters in NFT Markets with a Graph Learning System"
_ACM.org/TheWebConf/2024/Conference — TheWebConf24 Oral_

### Official Review · Reviewer_LPuG · 2023-11-04

**Novelty:** 6
**Technical Quality:** 6

**Review:**

This paper presents a study on the emerging issue of NFT Airdrop Hunters and proposes a novel graph representation learning technique integrated with multimodal features to detect potential hunters. The authors have utilized the data from NFT marketplace Blur for their evaluation, and their system, ARTEMIS, exhibits significant improvement over existing models in identifying hunters.

Generally, I like this paper, however, there are two major concerns that should be addressed before it can be accepted.

Pros:
- The paper addresses an interesting and relevant issue in the realm of NFTs - Airdrop Hunting. The topic is fresh and has not been extensively researched, making this study a significant contribution to the field.
- The proposed method is a creative combination of graph learning and multimodal features, designed to capture the nuanced behaviors of hunters. This technique is technically sound and innovative.
- The authors have conducted a comprehensive evaluation using data from the Blur market. The results demonstrate a substantial improvement over previous models, underscoring the effectiveness of the proposed system.

Cons:
- Concern one: The paper could benefit from a more detailed explanation of certain design decisions. The authors' rationale behind specific choices remains unclear (see more details in question)
- Concern Two: The paper does not clarify whether NFT Airdrop Hunting is a widespread issue across all marketplaces or is specific to certain ones. This information is crucial to understanding the scope and applicability of the proposed system.

**Questions:**

- The authors have utilized both the textual description and image attributes of NFTs to generate features. The ablation study indicates the usefulness of these multimodal features. However, it is unclear why these two features are significant in the context of Airdrop Hunters. In the threat model, these two attributes are defined by the vendor (victim) and should remain consistent across legitimate users and attackers (hunters). This suggests that these attributes may not be very useful in detecting hunters. I suspect that the multimode feature only serves as a unique identifier for **differentiating NFT assets**. If that is the case, using a multimodel instead of a simple hash ID seems to be overkill here. Please provide justification on why these advanced features are necessary in **differentiating attackers**.

- The evaluation of the system is based solely on data from one marketplace - Blur. There are numerous popular NFT marketplaces, such as OpenSea, Rarible, Mintable, Foundation, and Nifty Gateway. Is Airdrop Hunting a common problem across these marketplaces, or is it specific to Blur? The authors should clarify this to provide a better understanding of the system's applicability and effectiveness across different platforms.

**Reviewer Confidence:**

4: The reviewer is certain that the evaluation is correct and very familiar with the relevant literature

**Scope:**

4: The work is relevant to the Web and to the track, and is of broad interest to the community

---

### Official Review · Reviewer_2hfh · 2023-11-22

**Novelty:** 4
**Technical Quality:** 2

**Review:**

This paper introduces ARTEMIS, a graph neural network (GNN) system to detect airdrop hunters in NFT transactions. The main idea is to improve GNN sampling to consider NFT transaction paths and augment the model with metadata (text/image) embedding and price features. The system is evaluated on Blur’s data.

Strengths:

-	The proposed system is well explained, and the designs are reasonable.
-	The behavior analysis of airdrop hunters is informative.

Weaknesses:

-	Little information is provided on ground-truth labeling.
-	The detection performance is not good.
-	It is unclear what the text/image embedding is trying to achieve
-	Unclear if the detection method can be generalized to other airdrop applications

The system is largely reasonable by considering NFT metadata, NFT transactions, and anomalous trading prices in the modeling process. I have the following concerns.

**Data collection**

I don’t understand how the ground-truth labels are obtained. The paper stated that “we compared airdrop records to identify airdrop hunters meticulously. Subsequently, we sampled varying hunter scales and visualized microscopic transaction paths to validate data reliability” This is extremely vague and I cannot figure it out what is done here. Further details are needed on the ground-truth labeling procedure and the evidence of airdrop hunting.

**Feature Analysis**

For the hold time analysis, the paper argues the hold time of airdrop hunters is shorter (36 days) than regulars (53 days). I don’t think this is a super strong indicator given the large variance of the two distributions. Also, their median hold time seems to be quite similar based on the provided figure.

What are the image and text features supposed to catch? It would be helpful to explain the intuitions behind the embedding. Do you expect the airdrop hunters are go after those with similar text/words and images? Are these features trying to capture some form of “similarity”? Please clarify.

**Generalizability and Accuracy**

The system is designed based on a single dataset from Blur. It is unclear if the methodology can be generalized to airdrops at other platforms.

Unfortunately, after extensive feature engineering, the system performance is not satisfying. 0.71 precision and 0.729 recall is too low to deploy as a detector. The paper also lacks error analyses to understand what caused the misclassification.


Broken references:

- “Fig. 6 shows a detailed illustration of the process, which can be extended to multi-hop neighbors” – I don’t think fig6 is the right figure for this.
- “We tested the address distribution in the blur market and found it also follows a power-law distribution, with the results illustrated in fig. ??”

**Questions:**

Please see the above review.

**Reviewer Confidence:**

4: The reviewer is certain that the evaluation is correct and very familiar with the relevant literature

**Scope:**

3: The work is somewhat relevant to the Web and to the track, and is of narrow interest to a sub-community

---

### Official Review · Reviewer_2P8m · 2023-11-24

**Novelty:** 3
**Technical Quality:** 3

**Review:**

This paper introduces ARTEMIS, an optimized graph neural network system for identifying air-drop hunters in NFT transactions.

Pros: employs multi-modal attributes to enhance the detection performance of air-drop hunters.

Cons: The proposed method is a combination of existing methods.

**Questions:**

1. Why use ViT as the pre-trained transformer? Are there any potential alternatives?
2. The authors combine ViT and Bert in their method. This contribution has limited novelty.
3. The benchmarks used in the experiments are outdated.

**Reviewer Confidence:**

3: The reviewer is confident but not certain that the evaluation is correct

**Scope:**

3: The work is somewhat relevant to the Web and to the track, and is of narrow interest to a sub-community

---

### Official Review · Reviewer_VPCy · 2023-11-25

**Novelty:** 6
**Technical Quality:** 6

**Review:**

The authors built a system to detect airdrop hunters who create multiple accounts to profit from DApp token giveaways. This paper tackles an important and interesting research problem, but it needs several improvements.

The authors started by collecting transaction and airdrop data related to Blur. Clustering and labeling this data, they found 4,808 airdrop hunter wallet addresses. Unfortunately, no details were shared on how the clustering and labeling were performed and how accurate it was.

Then, they developed a complex architecture leveraging visual, textual, graph, and transaction-related features to train a model composed of VIT, BERT, GNN, and other NN components.

The model trained by the authors outperforms baseline models. While the authors performed some hyperparameter tuning for ARTEMIS, they don’t discuss such efforts for the baseline models. The lack of tuning of baseline models suggests that the comparison was potentially unfair.

While the model outperformed the best baselines, its precision is only 0.71, meaning the classifier will make 3 FP decisions for every 7 TP classifications. Based on how active airdrop hunters are, these false positives would mean that some of the most active benign users could be affected if ARTEMIS is used in real life, potentially negatively impacting the DApp users. It would have been great if the authors included a study about the FPs.

It would make the paper better if the authors included an adversarial discussion on how hard/easy it would be for airdrop hunters to evade ARTEMIS.

**Questions:**

How did you cluster and label airdrop hunters? How did you validate that an address is as an airdrop hunter?

Could a DApp set up a token giveaway that doesn’t incentivize airdrop hunting?

How much effort did you make to improve the baseline models (e.g., hyperparameter tuning? Do you think this comparison was far?

How easy would it be for airdrop hunters to adapt to your model and avoid detection?

How could Blur utilize Artemis without hurting benign users, given that ARTEMIS only achieved 0.71 precision?

**Reviewer Confidence:**

3: The reviewer is confident but not certain that the evaluation is correct

**Scope:**

3: The work is somewhat relevant to the Web and to the track, and is of narrow interest to a sub-community

---

### Official Review · Reviewer_MYPg · 2023-11-27

**Novelty:** 3
**Technical Quality:** 3

**Review:**

The study focuses on detecting Airdrop Hunters within the NFT trading domain in Web3. It introduces ARTEMIS as a solution with three key components. Firstly, the system employs a tailored neighbor sampling method and aggregator to connect multi-hop NFT transaction sequences. Secondly, ARTEMIS incorporates modules for multimodal feature extraction - images and descriptions. These modules leverage Transformer-based pre-trained models to extract both visual and textual insights from NFTs. Thirdly, the system engineers the representation of common NFT prices and advanced features focused on hunters, drawing from market manipulation theories and domain knowledge.
Strength:
1. The authors gather a real-world dataset which can be used for future studies for related research.
2. The idea of using transaction-based paths for node sampling is well designed, and based on their claim it is the first work of using ML in airdrop hunting.
3. The evaluation shows the effectiveness of the model for the problem compared to other graph-based and non-graph methods.

Weakness:
1. Except the novelty of problem, the core of the mythology doesn't look novel regarding graph learning and NFT feature extraction.
2. The presentation of the paper can be further improved regarding the attack model of airdrop hunter, and the necessities of using ML model to detect hunters.
3. Experiments are over simplified. The characteristics of the the patterns of airdrop hunters is not demonstrated.
4. There are some minor problems in the text. For example, on page 7, fig ?? should be corrected. Also, line 49.

**Questions:**

1. The code and the link to the dataset are missing in the paper. Are the authors planning to open source them?
2. The model uses attention for fusion. Can authors interpret the attention distribution for the text and image to clarify which part has made the most contribution?
3. Are there other datasets that can be used to further investigate the results?
4. What is the attack model of airdrop hunter
5. Based on the Figure 2, it looks like the airdrop hunters usually open multiple accounts and make fake transactions among them, leading to the creation of cycles in user graphs. Can this problem solved by cycle detection in the graph? Why is ML method the good way to address this problem?

**Reviewer Confidence:**

3: The reviewer is confident but not certain that the evaluation is correct

**Scope:**

3: The work is somewhat relevant to the Web and to the track, and is of narrow interest to a sub-community

---

### Decision · Program_Chairs · 2024-01-22

**Decision:**

Accept (Oral)

**Comment:**

While the proposal might suffer from some over-simplification, such as: little information provided about ground-truth labeling and a need for further refinement of the proposed methodology (the detection performance is not particularly good), the addressed problem is novel and there is convergence on the fact that the quality of the technical analysis is very high (combining solutions that, per se, already exist).
 There has been an active and useful exchange between the authors and reviewers.

 ---